# Feline Infectious Peritonitis Effusion Index: A Novel Diagnostic Method and Validation of Flow Cytometry-Based Delta Total Nucleated Cells Analysis on the Sysmex XN-1000V^®^

**DOI:** 10.3390/vetsci11110563

**Published:** 2024-11-13

**Authors:** Ricardo Lopes, Filipe Sampaio, Hugo Lima de Carvalho, Andreia Garcês, Cátia Fernandes, Carolina Vitória Neves, Alexandre Sardinha de Brito, Tiago Marques, Carlos Sousa, Ana Rita Silva, Ângela Martins, Luís Cardoso, Ana Cláudia Coelho, Elsa Leclerc Duarte

**Affiliations:** 1Department of Veterinary Sciences, University of Trás-os-Montes e Alto Douro (UTAD), 5000-801 Vila Real, Portugal; lcardoso@utad.pt (L.C.); accoelho@utad.pt (A.C.C.); 2Department of Veterinary and Animal Sciences, University Institute of Health Sciences (IUCS), CESPU, 4585-116 Gandra, Portugal; 3CEDIVET Veterinary Laboratories, Lionesa Business Hub, R. Lionesa 446 C24, 4465-671 Leça do Balio, Portugal; filipe.sampaio@cedivet.pt (F.S.); hugo.carvalho@cedivet.pt (H.L.d.C.); carolina.neves@cedivet.pt (C.V.N.); alexandresardbrito@gmail.com (A.S.d.B.); 4Cytology and Hematology Diagnostic Services, Laboratory of Histology and Embryology, Department of Microscopy, ICBAS-School of Medicine and Biomedical Sciences, University of Porto (U.Porto), Rua de Jorge Viterbo Ferreira, 228, 4050-313 Porto, Portugal; 5Wildlife Rehabilitation Centre (CRAS), Veterinary Teaching Hospital, University of Trás-os-Montes e Alto Douro (UTAD), 5000-801 Vila Real, Portugal; andreiamvg@gmail.com; 6Animal and Veterinary Research Centre (CECAV), Associate Laboratory for Animal and Veterinary Sciences (AL4AnimalS), University of Trás-os-Montes e Alto Douro (UTAD), 5000-801 Vila Real, Portugal; angela@utad.pt; 7Anicura Santa Marinha Veterinary Hospital, R. Dom Henrique de Cernache 183, 4400-625 Vila Nova de Gaia, Portugal; catia.fernandes@anicura.pt; 8Infectious Diseases Department, Santa Maria Hospital, Northern Lisbon University Hospital Centre (CHULN), 1649-035 Lisboa, Portugal; tiagomarques@medicina.ulisboa.pt; 9Molecular Diagnostics Laboratory, Unilabs Portugal, Centro Empresarial Lionesa Porto, Rua Lionesa, 4465-671 Leça do Balio, Portugal; carlos.sousa@unilabs.com (C.S.); ana.rita.silva@unilabs.com (A.R.S.); 10Department of Veterinary Medicine, School of Science and Technology, University of Évora, Polo da Mitra, Apartado 94, 7002-554 Évora, Portugal; emld@uevora.pt; 11Mediterranean Institute for Agriculture, Environment and Development (MED), Global Change and Sustainability Institute (CHANGE), University of Évora, Polo da Mitra, Apartado 94, 7002-554 Évora, Portugal

**Keywords:** clinical pathology, coronavirus, delta total nucleated cells (∆TNC), effusions, feline infectious peritonitis (FIP), flow cytometry, FIP Effusion Index, One Health, Stockholm paradigm, zoonosis

## Abstract

Feline infectious peritonitis (FIP) is a severe and even fatal inflammatory disease of cats, caused by feline coronavirus (FCoV), and remains an important diagnostic challenge. This study validates the flow cytometry-based delta total nucleated cells (∆TNC) on the Sysmex XN-1000V^®^ in effusion samples and introduces the FIP Effusion Index, a novel diagnostic method that integrates cellular and inflammatory markers of FIP in effusions. The FIP Effusion Index demonstrated improved diagnostic accuracy, achieving 96.3% sensitivity and 95.7% specificity for values of 5.06 or higher, and perfect specificity (100%) with 96.3% sensitivity at values of 7.54 or above. This combined approach outperforms traditional methods, providing a superior, rapid, and cost-effective tool for the accurate and timely FIP diagnosis. It offers considerable potential for enhancing the management of suspected FIP cases in clinical practice, with potential applications in both veterinary and human medicine for related coronavirus diseases.

## 1. Introduction

Coronaviruses (CoVs) constitute a large family of RNA viruses that infect a broad range of mammals including humans avian species, livestock and companion animals. Some cause severe disease, which is significantly challenging to public health, veterinary medicine, and the global economy [1,2,3,4,5]. The high diversity of these viruses facilitates the emergence of new variants with altered target cell preferences or expanded host ranges [6]. Such adaptations can result in cross-species transmission and zoonotic outbreaks. Recent examples of the latter are severe acute respiratory syndrome coronavirus (SARS-CoV), Middle East respiratory syndrome coronavirus (MERS-CoV), and the most recent severe acute respiratory syndrome coronavirus 2 (SARS-CoV-2) causing coronavirus infectious disease 2019 (COVID-19) [7,8,9,10,11,12].

The COVID-19 pandemic has highlighted the need for effective diagnostic model to better understand coronavirus-related diseases [13]. A notable characteristic of coronaviruses is their ability to alter tissue or cell tropism. The best prototypic example of such evolutionary adaptation is the feline coronavirus (FCoV), which illustrates how a virus can shift its pathogenic profile within a host or between hosts [8,12,13,14].

Feline infectious peritonitis (FIP), caused by FCoV, is a widespread and considerably fatal, severe systemic inflammatory disease in cats, caused by a mutated FCoV and its interaction with the host’s immune response [15]. Although SARS-CoV-2 and FCoV belong to different genera within the *Coronaviridae* family—FCoV to *Alphacoronavirus* and SARS-CoV-2 to *Betacoronavirus*—both exhibit rapid disease progression and diverse extrapulmonary manifestations [16,17]. Cats are susceptible to SARS-CoV-2, and FCoV has also been shown to infect humans [12,18,19,20,21]. These similarities suggest that FIP could serve as a valuable model for studying coronavirus-induced pathology, potentially framing diagnostic strategies in both veterinary and human medicine [7,16,21,22,23,24,25,26].

FCoV strains are subdivided into two distinct biotypes: feline enteric coronavirus (FECV) and feline infectious peritonitis virus (FIPV). The FIPV biotype is more virulent than the FECV biotype and can cause peritonitis with a poor prognosis, while most FECV biotypes do not cause lesions [18,27,28]. Serotype 1 FCoV includes unique feline strains, while serotype 2 appears to have arisen from the recombination between type 1 FCoV and canine coronavirus (CCoV). Although serotype 1 is the most prevalent worldwide, serotypes 1 and 2 can cause FIP. In cats with FIP, the virus replicates to high titres in monocytes and can be found in multiple organs. In subclinical infected cats, on the other hand, FCoV is mainly confined to the intestine. Even pathogenic strains of the FECV biotype can cause only mild enteritis because of their very low virulence [18,29,30,31,32].

The pathogenesis of FIP involves complex interactions including two amino acid substitutions (M1058L and S1060A) in the spike protein of the virus, which are associated with the systemic spread of FCoV rather than the direct causation of FIP. These mutations highlight the diagnostic challenge, as current tests are unable to properly differentiate between enteric and pathogenic FCoV strains, emphasising the need to improve diagnostic tools for this purpose, which offer higher sensitivity and specificity [27,29,33,34,35,36,37].

Diagnosing FIP antemortem presents considerable challenges, particularly in the non-effusive (“dry”/parenchymatous) form, due to its variable clinical manifestations and the low specificity of many laboratory tests. The effusive (“wet”/non-parenchymatous) form of FIP is easier to diagnose because it presents more clearly through patient history, clinical signs, and the analysis of effusions, which are typically yellow, turbid, viscous, and often contain fibrin strands [30,38,39,40].

It is worth mentioning that, although the distinction between effusive and non-effusive forms may provide some assistance in the diagnostic approach, they do not represent two distinct entities of FIP. It is common for cats that initially present with the non-effusive form to eventually develop effusions in the later stages of the disease [30,41].

These effusions typically have a high protein content with a decreased albumin-to-globulin (ALB/GLOB) ratio. Cell counts range from 2–6 × 10^3^/μL, sometimes up to 30 × 10^3^/μL, and the cytologic pattern, which is only highly suggestive but not definitive diagnostic for FIP, mostly consists of nondegenerate neutrophils, macrophages, lymphocytes, and rare plasma cells in a prominent proteinaceous background [42,43].

Current diagnostic methods, such as cytological analysis, real-time quantitative reverse transcription polymerase chain reaction (real-time RT-qPCR), serum protein electrophoresis, alpha-1-acid glycoprotein (AGP) measurement, FCoV antigen detection, and others, especially when combined, can strongly suggest FIP but do not provide a definitive diagnosis [28,38,42,44]. Despite innovations in the field, an ideal diagnosis for FIP still does not exist. Detection of the FCoV antigen by immunohistochemistry (IHC) on histopathological abnormal tissue obtained either post–mortem or via laparotomy/laparoscopy remains the gold standard of diagnosis [38,40].

PCR is a highly sensitive and specific technique for detecting DNA or RNA sequences, making it effective for diagnosing infectious diseases including coronaviruses, where it is commonly used as part of routine testing [45]. RT-qPCR is particularly useful for RNA viruses like FCoV, as it converts RNA to complementary DNA (cDNA) and amplifies specific sequences in real-time, allowing for the rapid detection and quantification of viral load, which is crucial for early FIP diagnosis and management [46,47,48]. However, RT-qPCR’s high sensitivity and specificity come with considerable costs for specialised equipment, reagents, and skilled personnel, limiting its routine use in some veterinary practices. This highlights the need for more affordable and complementary diagnostic methods such as the Rivalta test, ALB/GLOB ratio, and delta total nucleated cell (∆TNC) count on effusions [30,36].

The Rivalta test, a simple and cost-effective assay, has gained recognition for its high diagnostic value in FIP. The test relies on the precipitation of proteins such as fibrinogen when in contact with acetic acid, a reaction particularly noticeable in FIP-related effusions [36,42,49]. However, this reaction is not exclusive to FIP and can also occur in cases of bacterial peritonitis, pleuritis, or lymphoma. Therefore, it is essential to combine the Rivalta test with cytological analysis to achieve a more accurate diagnosis [42,50].

The ALB/GLOB ratio is a practical and low-cost parameter for evaluating FIP effusions, providing valuable insights into the protein alterations associated with the disease. A decreased ratio suggests increased globulin and decreased albumin levels, indicating an inflammatory response typical of FIP and helping to differentiate it from other conditions [30].

Recent studies have highlighted the utility of combining tests with high positive likelihood ratios (LR+), such as the ΔTNC count on effusions, measured using the Sysmex XN-1000V^®^ analyser. This method offers an innovative approach by quantifying the difference in total nucleated cell counts between specific channels (TNCC-WDF and TNCC-WNR). This accuracy is attributed to the formation of clots in the WNR reagent, which trap cells similarly to the mechanism observed in the Rivalta test, a method also noted for its high diagnostic accuracy for FIP. When used on the Sysmex XN-1000V^®^, this method has demonstrated promising results comparable to those obtained with the Sysmex XT–2000iV^®^ [36,43].

The FIP Effusion Index integrates two independent variables, each providing unique but complementary insights into the nature of the effusion: the ∆TNC and the ALB/GLOB ratio. By integrating cellular and inflammatory responses with protein changes characteristic of FIP, the FIP Effusion Index aims to enhance diagnostic accuracy, reduce the likelihood of misdiagnosis, and provide a rapid and cost-effective approach for managing suspected FIP cases in clinical practice.

The present study aimed to address three primary objectives in accordance with the Standards for Reporting of Diagnostic Accuracy (STARD) guidelines [51]: (1) to determine the diagnostic accuracy of ∆TNC for FIP effusions using the Sysmex XN-1000V^®^, with real-time RT-qPCR as the reference method; (2) to compare the diagnostic accuracy of ∆TNC-XN with the Rivalta test and the ALB/GLOB ratio; and (3) to introduce a new diagnostic method, the FIP Effusion Index, which correlates ∆TNC and the ALB/GLOB ratio to minimise false-positive and false-negative outcomes, and evaluate its diagnostic accuracy in diagnosing FIP.

## 2. Materials and Methods

### 2.1. Data Collection, Sampling, and Diagnostic Procedures

Effusion samples from suspected cases of FIP were submitted to CEDIVET Veterinary Laboratories (Porto, Portugal). These samples were submitted from veterinary practices including clinics and hospitals from the north region of mainland Portugal. Along with the samples, a laboratory requisition was received, which included the complete clinical information for each cat, particularly their breed, sex, age, vaccination and prophylactic status, clinical suspicion/clinical signs, and requested analyses.

Although a large number of samples were submitted and all were fully analysed in accordance with the methods outlined below, only 50 samples fulfilled all the inclusion criteria. The cases selected for data analysis were based on the following inclusion and exclusion criteria:

Inclusion criteria:Only samples from cats with effusions reported by veterinary practitioners who were clinically suspected of FIP (e.g., lethargy, anorexia, weight loss, fever, jaundice, dyspnoea, abdominal lymphadenopathy, thickened intestinal walls, renomegaly, uveitis, vasculitis/phlebitis, retinal changes, multifocal neurologic signs) were included in this study.Availability of laboratory requisition with the relevant clinical information such as clinical suspicion/clinical signs and requested analyses.Availability of complete documentation of physicochemical analysis of the effusion including specific gravity (SG), protein content, absolute counts of white blood cells and total nucleated cells in body fluid, and total nucleated cells derived from the ratio between specific channel (TNCC-WDF and TNCC-WNR) counts on the Sysmex XN-1000V^®^.

Exclusion criteria:Absence of complete clinical documentation: Samples lacking a comprehensive laboratory requisition or clinical information including clinical suspicion/clinical signs and requested analyses were excluded.Lack of confirmed FCoV real-time RT-qPCR diagnosis: Cases without a confirmed diagnosis of FCoV through real-time RT-qPCR testing were excluded from the study.Incomplete physicochemical analysis data: Effusion samples without full documentation of physicochemical analysis such as SG, protein content, absolute counts of white blood cells and total nucleated cells in body fluid, and the total nucleated cells derived from the ratio between the TNCC-WDF and TNCC-WNR counts on the Sysmex XN-1000V^®^ were not included.Non-compliance with sampling protocols: Samples collected in non-EDTA tubes or samples submitted beyond the 18–24 h window post-collection were excluded to maintain the integrity of the cell count analysis.Samples with cytological findings that were septic, unclear, or inconclusive were excluded from the study.Previous treatments influencing effusion composition: Samples from cats that had undergone treatments known to affect the composition of effusions, such as recent chemotherapy or immunosuppressive therapy, were excluded to avoid confounding results.Insufficient sample volume: Effusions with insufficient volume to perform all required analyses were excluded to ensure that all parameters could be adequately measured.

Based on these criteria, cats were classified as affected by FIP if the effusion showed elevated protein levels (>2.5 g/dL), the cytological analysis was indicative of FIP, and a positive FCoV real-time RT-qPCR result was obtained [30,52,53,54].

The age of the animals was categorised into five groups: kitten, <1 year old; young, 1 to <2 years old; adult, 2 to <6 years old; senior, 6 to <11 years old; and old, ≥11 years old.

### 2.2. Biochemical Analysis

The measurements of total protein (TP) and albumin (ALB) were performed using the Beckman Coulter AU680e Automated Chemistry Analyzer^®^ (Beckman Coulter, Cassina De’Pecchi, Italy). Globulin levels were determined through indirect calculation. The effusion SG was measured using a clinical refractometer (Clinical Refractometer Mod. TS400; Sper Scientific, Scottsdale, AZ, USA).

### 2.3. Rivalta Test

To conduct the Rivalta test, 7–8 mL of distilled water was dispensed into a 10 mL plastic tube (Cat: 470332-708, VWR^®^, Radnor, PA, USA). A drop (20–30 µL) of acetic acid (98–100%) was then added using a disposable pipette (Merck, Darmstadt, Germany) and the mixture was thoroughly agitated. Subsequently, a drop (20–30 µL) of effusion fluid was carefully placed on top of the acetic acid solution with a second disposable pipette. The Rivalta test was deemed positive if a precipitate formed, adhered to the surface, retained its shape, or slowly settled to the bottom of the solution as a drop or jellyfish-like (Figure 1). A negative test was indicated by the drop dissipating (disappearing) and the solution remaining clear [49,55,56].

### 2.4. Flow Cytometry Analysis

The tests for the total white blood cell count in body fluid (WBC-BF), total nucleated cells in body fluid (TC-BF), total nucleated cells in the white blood cell differential channel in whole blood mode (TNCC-WDF), total nucleated cells in the white blood cell nucleated channel in whole blood mode (TNCC-WNR), and the ∆TNC ratio derived from the TNCC-WDF and TNCC-WNR counts were performed using the Sysmex XN-1000V^®^ (Sysmex Europe GmbH, Norderstedt, Germany) (Figure 2, Figure 3, Figure 4 and Figure 5). Confirmation of the effusion findings and cellular morphology was conducted through cytological evaluation using light microscopy, following staining with May-Grünwald Giemsa, on slides prepared by direct smear, squash preparation, and cytocentrifugation.

### 2.5. FIP Effusion Index

The FIP Effusion Index is calculated using Formula (1):(1)$$\mathrm{FIP}\;\mathrm{Effusion}\;\mathrm{Index}=\frac{\Delta \mathrm{Total}\;\mathrm{Nucleated}\;\mathrm{Cells}\;\left(\Delta \mathrm{TNC}\right)}{\mathrm{Albumin}/\mathrm{Globulin}\;\left(\mathrm{ALB}/\mathrm{GLOB}\right)\;\mathrm{ratio}},$$

The FIP Effusion Index is calculated by dividing the delta total nucleated cell (∆TNC) count by the albumin-to-globulin (ALB/GLOB) ratio. This formula integrates cellular and biochemical markers to enhance diagnostic accuracy in detecting FIP.

### 2.6. Effusion Classification

Effusions were classified according to the Traditional Classification Scheme [57,58], based on their protein concentration and total cell count. This scheme categorises effusions into transudate, modified transudates, and exudates, as exemplified in Table 1. The classification criteria include specific thresholds for protein levels and total cell counts, enabling a systematic approach to differentiating between types of effusions.

### 2.7. Molecular Analysis

The extraction of viral RNA from the effusion samples was conducted using the Promega AX9760 custom protocol for whole blood, applied through the KingFisher™ Flex system (Thermo Fisher Scientific, Waltham, MA, USA). The protocol was adapted for the extraction of nucleic acids from body fluid samples. All reagents were stable at room temperature and used under the manufacturer’s specified conditions, with storage between 15–30 °C. The target RNA needed to be isolated free from contaminants such as proteins, cellular debris, and non-target nucleic acids. To achieve this, the samples were first treated with lysis buffer and Proteinase K. After an incubation period of approximately 27 min, a binding buffer mix containing MagnaCell^TM^ Binding Buffer and paramagnetic resin was added. After the binding step, samples were washed with 80% ethanol to remove contaminants, and RNA was finally eluted using the elution buffer.

For the detection of FCoV in the effusion samples, molecular biology analyses were performed using the NZYTech Feline Coronavirus RT-qPCR Kit^®^ (NZYTech, Lisboa, Portugal), optimised to provide the broadest possible detection profile while maintaining specificity for FCoV, with over 95% homology to a wide range of FCoV genomes based on comprehensive bioinformatic analysis of reference data from the NCBI database. The real-time RT-qPCR reaction was configured according to the manufacturer. For each reaction, an internal extraction control was included. A negative control and a positive control were also included.

The amplification reaction was carried out on a QuantStudio™ 5 real-time PCR System (Thermo Fisher Scientific, Waltham, MA, USA), following the thermal cycling conditions suggested in the manufacturer’s protocol (Table 2).

During the amplification process, both primers and probes specifically annealed to a designated target region of the FCoV genome. The fluorogenic probe, consisting of a DNA sequence tagged with a fluorescent dye at the 5′ end and a quencher molecule at the 3′ end, was cleaved during the PCR amplification process. This cleavage caused the dye and quencher to separate, leading to an increase in fluorescence intensity. The rise in fluorescence was subsequently measured by the instrument, indicating successful target amplification.

For the final validation of the results, the following criteria had to be met:Amplification of the internal extraction control;No amplification of the negative control;Amplification of the positive control.

Only results meeting these criteria were validated as negative (no amplification of the target), positive (true amplification of the target, indicated by a sigmoidal curve), or invalid (if there was no amplification of the target and no amplification of the internal extraction control, and therefore excluded from the study).

### 2.8. Statistical Analysis

All the data were available in digital format in Sislab^®^ version 23R12.1.1.1 (Glintt, Global Intelligent Technologies, S.A., Lisboa, Portugal) and transferred to Microsoft Excel^®^ version 2410 (Microsoft, Redmond, WA, USA) sheets. Statistical analysis was conducted using the JMP^®^, version 14.3 (SAS Institute, Cary, NC, USA, 1989–2023), DATAtab^®^ (DATAtab: Online Statistics Calculator. DATAtab e.U. Graz, Austria, 2024), and MedCalc^®^ Statistical Software version 20.006 (MedCalc Software Ltd., Ostend, Belgium, 2021). Non-parametric tests were employed to study the differences between the observed and expected frequencies of categories within a field including the binomial test, the one-sample Chi-square test, the chi-square test of independence, and Fisher’s exact test, depending on the number of categories in the categorical field. For comparisons among three or more independent groups, the Kruskal–Wallis test was used, followed by the Dunn–Bonferroni post hoc test for multiple comparisons when appropriate. The Cochran–Armitage trend test was also used to assess trends across ordered categories. Odds ratios (ORs) were calculated to evaluate the likelihood of a positive Rivalta test in cats with FIP compared to those without, providing a measure of association between the presence of FIP and the test outcomes. To evaluate the diagnostic accuracy of the ALB/GLOB ratio, and the ∆TNC count, derived from the ratio of TNCC-WDF to TNCC-WNR on the Sysmex XN-1000V^®^, a mixed-model ANOVA was performed on the biochemical and flow cytometry data, and receiver operating characteristic (ROC) curve analysis was utilised to further assess the diagnostic effectiveness of ∆TNC and the FIP Effusion Index in identifying FIP in the effusion samples. The sample parameters were categorised into two groups:

Qualitative variables: These involved the breed, sex and age categories (kitten: <1 year old; young: 1 to <2 years old; adult: 2 to <6 years old; senior: 6 to <11 years old; old: ≥11 years old) as well as the results of the Rivalta test and effusion classification.

Quantitative variables: These involved biochemical parameters such as TP, ALB, GLOB, and ALB/GLOB ratio. The physical parameter, effusion SG, was also included. Flow cytometry data included the WBC-BF, TC-BF, TNCC-WDF, TNCC-WNR, and ∆TNC derived from the TNCC-WDF and TNCC-WNR counts. Additionally, the FIP Effusion Index, which integrates the ALB/GLOB ratio and ∆TNC, and the quantitative molecular analysis results from the Feline Coronavirus RT-qPCR Kit^®^, classified in accordance with the manufacturer’s test specifications, were included.

## 3. Results

### 3.1. Qualitative Variables

#### 3.1.1. Descriptive Data

Of the 50 animals included in this study, 23 (46%; 95% CI: 33.0–59.6%) tested negative with the Feline Coronavirus RT-qPCR Kit^®^ while 27 (54%; 95% CI: 40.4–67.0%) tested positive.

#### 3.1.2. Breed

Regarding breed, our study population comprised animals from six different cat breeds that included the following: 42 Domestic Shorthairs (84%), three Scottish Fold (6%), two Persian (4%), one Sphynx (2%), one British Shorthair (2%), and one Siamese (2%).

The Kruskal–Wallis test indicated no significant association between the cat breed and the negative and positive outcomes of the coronavirus molecular biology test in effusions (*p* = 0.337), suggesting no variability in FIP frequency across breeds. Detailed ranks and pairwise comparisons were performed, revealing specific breed differences, but none reached significance after adjustment for multiple comparisons. The Dunn–Bonferroni post hoc test revealed no significant associations, with all adjusted *p*-values exceeding the 0.05 significance level.

#### 3.1.3. Sex

From the 50 animals analysed, 32 (64%) were males and 18 (36%) were females. Table 3 represents the percentages of positive and negative tests according to sex. The differences observed between animal sexes were not significant (*p* = 0.174).

#### 3.1.4. Age

From the 50 animals analysed, the age distribution ranged from ≤1 year (6 months) to 17 years, with an average age of 4.7 ± 3.9 years. A total of 26% (95% CI: 15.9–39.6%; *n* = 13) were kittens, 24% (95% CI: 14.3–37.4%; *n* = 12) were young, 26% (95% CI: 15.9–39.6%; *n* = 13) were adults, 14% (95% CI: 7.0–26.2%; *n* = 7) were seniors, and 14% (95% CI: 4.3–21.4%; *n* = 5) were old.

Table 4 displays the occurrence of coronavirus identification according to age group. The Spearman correlation analysis indicated no significant relationship between age groups and the frequency of FIP, as evidenced by a *p*-value of 0.083. Therefore, it cannot be concluded that age influences the frequency of feline coronavirus infection among the groups studied.

#### 3.1.5. Rivalta Test Results

Regarding the Rivalta test, 33 (66%) were positive and 17 (34%) were negative. The analysis revealed a significant association between the Rivalta test and FCoV real-time RT-qPCR results, confirmed by Fisher’s exact test (*p* < 0.001). This trend is further supported by the Cochran–Armitage test, highlighting a significant association between the two parameters (*p* < 0.001), suggesting that feline infectious peritonitis effusions are more likely to have a positive Rivalta test. The sensitivity of the Rivalta test, which is its ability to correctly identify cats with FIP, was approximately 96.3%. The specificity, or the test’s ability to correctly identify cats without the disease, was approximately 69.6%. The odds ratio (OR) for a cat with FIP having a positive Rivalta test in the effusion was 59.4 (95% CI: 6.68–528.82), indicating that cats with FIP were 59 times more likely to have a positive Rivalta test compared to cats that were negative for FCoV real-time RT-qPCR in the effusion.

Table 5 represents the percentages of positive and negative Rivalta tests according to the FCoV real-time RT-qPCR.

#### 3.1.6. Effusion Classification

Among the effusion samples analysed, 46 (92%) were classified as modified transudates, while 4 (8%) were exudates. No pure transudates were identified. Statistical analysis revealed no correlation between effusion type and the FCoV real-time RT-qPCR results (*p* = 0.87). Table 6 represents the percentages of effusion type according to the FCoV RT-qPCR.

The cytological features of effusions associated with FIP observed in the present study included fluid that was often clear and yellow, with a sometimes viscous consistency. The fluid was occasionally mildly haemodiluted. Cellular density varied widely but was generally considered low. The background of these samples was noted to be pale to moderately basophilic and granular, frequently containing numerous protein crescents. The nucleated cells identified were usually a mix of non-degenerate neutrophils, monocytes/macrophages, and small lymphocytes (Figure 6).

### 3.2. Quantitative Variables

#### 3.2.1. Biochemical and Flow Cytometry Data

Spearman’s correlation analysis (Appendix A) was performed on the data obtained. Table 7 presents the values, in effusions, of the biochemical and flow cytometry data WBC-BF, TC-BF, TNCC-WDF, TNCC-WNR, and ∆TNC, categorised according to the Feline Coronavirus RT-qPCR Kit^®^ test results, showing comparisons between cats with either negative or positive outcomes.

To facilitate interpretation, the authors chose to expound the most salient associations. The Spearman correlation analysis showed a significant correlation (*p* < 0.001) between the ALB/GLOB ratio and the FCoV RT-qPCR test results. Similarly, our study identified a strong positive correlation (*r* = 0.85, *p* < 0.001) between the ∆TNC values and FCoV real-time RT-qPCR results, suggesting that higher ∆TNC values are associated with positive test outcomes.

The ∆TNC was significantly higher (*p* < 0.001) in cats with FIP (median: 27.0; min-max: 1.04–583; 95% CI: 14.7–102.2) than in non-FIP cats (median: 0.9; min-max: 0.31–4.90; 95% CI: 0.8–1.8), and the TNCC-WNR and TNCC-WDF counts were significantly higher (*p* < 0.001 and *p* < 0.05, respectively) in cats with FIP (TNCC-WNR = 0.3; 0.0–3.89; TNCC-WDF = 2.32; 0.06–16.75) than in non-FIP cats (TNCC-WNR = 24,570.9; 0.05–342,445; TNCC-WDF = 24,398; 0.1–335,752) (Figure 7). Results from these latter cats were characterised by a high interindividual variability, likely due to the heterogeneity of the diseases responsible for the effusions. All of the cats with FIP had a ∆TNC ≥ 2.1, except for one cat that had “atypical” FIP (∆TNC 1.04). All non-FIP cats had a ∆TNC < 4.9. More specifically, only 2 specimens from all 23 cats without FIP had a ∆TNC ≥ 2.1.

#### 3.2.2. Diagnostic Accuracy of the ∆TNC

A mixed-model ANOVA was utilised to evaluate the biochemical and flow cytometry data, categorising them based on their responses to the FCoV RT-qPCR Kit^®^ test results as either negative or positive.

The analysis highlighted strong associations of ∆TNC levels in positive cases with statistical significance (*p* < 0.001), suggesting that the total nucleated cells derived from the ratio between the TNCC-WDF and TNCC-WNR counts on the Sysmex XN-1000V^®^ dynamics were notably influenced by the FCoV infection status (Figure 8).

#### 3.2.3. Receiver Operating Characteristic (ROC) Curve Analyses

The area under the ROC-AUC for ALB/GLOB ratio in the effusions was 0.968 (σ = 0.024; 95% CI: 87.4–99.7%), indicating excellent discriminatory power (*p* < 0.0001). The Youden index was 0.9195, with an associated criterion of ALB/GLOB ratio ≤ 0.56, yielding a sensitivity of 96.30% and a specificity of 95.65%.

The analysis of the ∆TNC using the Sysmex XN-1000V^®^ in feline effusions for diagnosing FIP demonstrated significant diagnostic efficacy. Among the 50 samples analysed, 27 (54%) were from cats with a positive FCoV RT-qPCR Kit^®^ test result, and 23 (46%) were from cats with a negative result. The ROC-AUC for ∆TNC was 0.990 (σ = 0.009; 95% CI: 91.1–100%), indicating excellent discriminatory power (*p* < 0.0001) (Figure 9 and Table 8). The Youden index was 0.9259, with an associated criterion of ∆TNC ≥ 4.903, yielding a sensitivity of 92.59% and a specificity of 100%.

#### 3.2.4. FIP Effusion Index

The ROC-AUC for the FIP Effusion Index was 0.995 (σ = 0.006; 95% CI: 92.0–100%), indicating excellent discriminatory power (*p* < 0.0001). For an FIP Effusion Index ≥ 5.06, the sensitivity was 96.30% and the specificity was 95.65%. Using an optimal criterion determined by the Youden index of 0.9630, a FIP Effusion Index ≥ 7.54 achieved a sensitivity of 96.30% and a specificity of 100% (Figure 10 and Table 9).

## 4. Discussion

CoVs are highly adaptable, with significant genetic diversity that allows for the emergence of new variants that are capable of cross-species transmission, leading to outbreaks like SARS-CoV, MERS-CoV, and SARS-CoV-2, which caused the COVID-19 pandemic. FCoV, responsible for FIP, exemplifies this adaptability, ranging from asymptomatic infections to severe systemic disease. This variability within the same host species makes FCoV an excellent model for studying coronavirus evolution and pathogenesis. FIP is particularly challenging to diagnose due to the lack of pathognomonic signs and its varied presentations within the same individual, highlighting the need for reliable diagnostic tools [3,4,5,9,10,11,13,14,16,17,19,20,21,25,39,59].

The current guidelines [30,34,38,60] for FIP diagnosis highlight the complexities of FIP diagnosis, emphasising the need for understanding the sensitivity, specificity, and predictive values of diagnostic tests, especially when no effusion is present. Guidelines suggest that building an index of suspicion ‘brick by brick’ through a combination of history, signalment, and targeted diagnostic tests is crucial. Given the fatal nature of untreated FIP, achieving a correct diagnosis is paramount [61,62,63,64].

The present study addressed these challenges by both developing the FIP Effusion Index, a novel diagnostic tool that integrates biochemical and cellular markers to enhance diagnostic accuracy and reduce misdiagnosis and validating the flow cytometry-based ∆TNC on the Sysmex XN-1000V^®^ in effusion samples.

The limited sample size of 50 individuals, dictated by stringent inclusion and exclusion criteria, was a constraint. Future studies should include a larger cohort, incorporating IHC FCoV antigen detection and including septic cases to further validate these findings and explore the index’s applicability in monitoring patients with persistent effusions.

### 4.1. Sex, Breed and Age

The findings of this study regarding breed, sex, and age offer a refined perspective on the diagnosis of FIP. Although purebred cats are often regarded as being at a higher risk for FIP, our results indicate that Domestic Shorthairs, the most frequent breed in this study, still represented a considerable proportion of cases, indicating the extensive reach of FIP across all breeds. This observation is consistent with previous studies [33,65,66] and contrary to the expectations set by other published studies [67,68], which often suggest that purebred cats are more likely to develop FIP. However, the lower frequency of FIP in purebred cats within the present study may be attributed to the specific population demographics of cats presented at the participating clinics or even to sampling bias. Concerning sex, while some studies have indicated a marginally higher prevalence of FIP in male cats, potentially due to hormonal influences or behavioural traits, this evidence remains inconclusive [33,69]. Our findings are consistent with this ambiguity, suggesting that sex does not constitute a definitive risk factor for FIP in the studied population. No significant correlation was found between age and the occurrence of FIP. This result contrasts with the expectations set by many published studies [30,33,34,65,70], which suggest that younger cats, particularly those under 2 years of age, are more vulnerable to FIP.

### 4.2. Effusion Classification

The most common clinical presentation in cats with FIP is the presence of effusion [30]. Typically, FIP effusions contain high protein concentrations and low white blood cell counts and therefore may be classified as either modified transudates or exudates, depending on the total protein concentration and total cell count. However, some cats with FIP can have very high cell counts in the effusion (e.g., secondary bacterial peritonitis) [30,38,71].

Although the vast majority of cats with FIP will present with effusion, diagnostic algorithms for animals with this clinical presentation should consider various pathophysiological mechanisms (or a combination of them) such as altered hydrostatic and oncotic pressures, increased vascular permeability to plasma proteins, leakage of blood from vessels, leakage of lymph from lymphatic vessels, and rupture of hollow organs or tissues to determine the primary diagnosis [53,72].

Another study [65] reviewed the medical records of 231 cats with confirmed FIP and found that 78.1% (175/224) of the cats evaluated had effusion. In contrast, a different study [54] examined a more extensive sample of cats with thoracic and abdominal effusions using cytological criteria and refractometric estimates of the total protein concentration. In this study, only 9.6% (38 out of 396) of the cases were suggestive of FIP, highlighting the range of other potential causes of effusion in felines such as septic, haemorrhagic, neoplastic, and other conditions.

In our study, a large portion of the samples (86%) were classified as modified transudates. Although classifying effusions is essential in clinical and laboratory contexts, this classification alone does not provide a definitive diagnosis. The classification of effusions is primarily used to guide or provide evidence for the type of pathological process that led to the effusion, which helps identify the primary diagnosis [53].

When statistically correlating the type of effusion with the FCoV RT-qPCR test results, no statistical significance was observed. Although FIP is considered a common cause of modified transudates, many of the effusions analysed in this study and classified as modified transudates were not consistent with FIP.

The absence of statistical significance in this context supports the idea that in the presence of an effusion with a clinical suspicion of FIP and with a cytological classification of modified transudate, other differential diagnoses should be also considered. According to some authors [30,53,60], traditional laboratory data, such as protein concentration and cytological evaluation, are often insufficient to diagnose the underlying disorder. In such cases, additional information including patient history, physical examination, imaging findings, or other tests is necessary to accurately interpret the significance of laboratory data and establish the pathogenesis of an effusion. Thus, the absence of statistical significance between modified transudate and FIP further emphasises the need for additional data and techniques to refine the diagnosis, as demonstrated in our study with the FIP Effusion Index or other methods such as immunocytochemistry, PCR, Rivalta, and others.

### 4.3. ∆TNC-XN, Rivalta Test, and ALB/GLOB Ratio in FIP Diagnostic Approaches

Currently, there are various diagnostic approaches for FIP including cytological analysis, real-time RT-qPCR, serum protein electrophoresis, and the Rivalta test, among others. While a conclusive diagnosis cannot be established, these methodologies, when employed together, may provide strong evidence supporting the presence of FIP. According to the literature review by the authors [15,38], real-time RT-qPCR has a demonstrated high sensitivity and specificity, often reaching 100% in several studies [30,73,74,75,76,77,78,79,80,81] when compared to the detection of the FCoV antigen by IHC. Therefore, although it is still considered the gold standard, real-time RT-qPCR was selected as the standard test for this study due to its non-invasive nature, speed of results, and suitability for integration into clinical routine.

The Rivalta test is well-known for its simplicity and cost-effectiveness. It demonstrates high sensitivity (96.3%) in identifying FIP-related effusions but has a moderate specificity (69.6%), which results in a higher proportion of false positives [49,50]. This can lead to the overdiagnosis of FIP, particularly in cases of bacterial peritonitis, pleuritis, or lymphoma, where similar effusion characteristics might be present [49,50].

In contrast, the ALB/GLOB ratio in effusions, a biochemical marker, showed excellent diagnostic accuracy with an ROC-AUC of 0.968, sensitivity of 96.3%, and specificity of 95.7% for a criterion of ALB/GLOB ratio ≤ 0.56. This indicates that the ALB/GLOB ratio effectively distinguishes between FIP and non-FIP effusions, outperforming the Rivalta test in specificity and thereby reducing the number of false positives.

The ∆TNC approach, analysed through the Sysmex XN-1000V^®^ flow cytometry, in FIP effusions, demonstrates significant advantages over traditional diagnostic methods. This method outperformed both the Rivalta test and the ALB/GLOB ratio in terms of specificity and presented a near-perfect diagnostic tool. The ability of ∆TNC to effectively differentiate FIP from other effusion-causing conditions, such as inflammation or neoplasms, underlines its utility as a superior diagnostic marker in veterinary practice. In the present study, the ∆TNC method showed high sensitivity (92.6%) and specificity (100%) in detecting FIP-associated effusions, with a ∆TNC value above 2.1 indicating a likely FIP diagnosis and values above 4.9 may be considered diagnostic for FIP. Post hoc Bonferroni tests further supported these results, showing that the total cell count dynamics significantly differed between groups, underscoring the potential of ∆TNC as a sensitive biomarker for diagnosing FCoV infection. These findings align with previous studies [42,43] using similar methodologies, such as the Sysmex XT-2000iV^®^, reinforcing the robustness and reliability of this innovative approach in clinical practice. Thus, ∆TNC has emerged as a highly dependable marker for distinguishing FIP-associated effusions, with near-perfect accuracy in identifying true positive and true negative cases, underscoring its value in the diagnostic evaluation of suspected FIP in cats [42].

### 4.4. FIP Effusion Index: A Novel Diagnostic Method

The FIP Effusion Index improves diagnostic accuracy by combining the ALB/GLOB ratio with the ∆TNC for FIP for several reasons. First, the ALB/GLOB ratio and the ∆TNC provide complementary diagnostic information regarding the biochemical and cellular characteristics of effusions. The ALB/GLOB ratio serves as an indirect indicator of specific protein proportions within the fluid, with a decreased ALB/GLOB ratio typically observed in FIP-associated effusions due to increased globulin and decreased albumin levels. Meanwhile, the ∆TNC, obtained via flow cytometry, measures changes in the nucleated cell population, serving as an indirect marker of inflammatory cellular response burden within the effusion. By combining these two measures, the FIP Effusion Index captures both biochemical and cellular alterations, offering a more comprehensive and integrated diagnostic evaluation. This method outperformed the Rivalta test, ALB/GLOB ratio, and ∆TNC individually, underscoring the FIP Effusion Index as a superior method for distinguishing FIP from other conditions with similar effusion characteristics, making it an almost ideal diagnostic tool for FIP.

Furthermore, the use of flow cytometry-based ∆TNC analysis with the FIP Effusion Index offers a less invasive and quicker alternative to molecular diagnostic methods like real-time RT-qPCR. Although PCR has a high diagnostic value, it requires longer processing times and is more resource-intensive. The comparable diagnostic effectiveness of the FIP Effusion Index to real-time RT-qPCR further supports its inclusion in standard diagnostic protocols. With its high sensitivity and specificity, the FIP Effusion Index is a valuable tool in veterinary diagnostics, particularly in differentiating FIP from other causes of effusions in cats.

The study’s adherence to the Standards for Reporting of Diagnostic Accuracy (STARD) guidelines ensures that the findings are robust and applicable to larger populations. The optimal cutoff value for ∆TNC and the FIP Effusion Index that minimises false positives and negatives has been clearly defined, providing a practical guideline for clinicians, as demonstrated in the diagnostic algorithm presented in Figure 11. This is particularly important in veterinary practices, where distinguishing FIP from similar conditions is challenging due to overlapping clinical signs and effusion characteristics [51].

The FIP Effusion Index analysis using the Sysmex XN-1000V^®^ represents a significant advancement in diagnosing FIP in cats, offering high accuracy, practicality, and efficiency, which positions it as a superior alternative to traditional methods. This study supports the integration of the FIP Effusion Index into routine veterinary diagnostics for FIP, providing a reliable tool that improves diagnostic accuracy and clinical decision-making.

The use of Sysmex XN-1000V^®^’s channels, specifically WNR and WDF, introduces an innovative technique for differentiating between FIP-positive and FIP-negative cases. The reagents used in these channels differ from those in the analogous BASO and DIFF channels of the Sysmex XT-2000iV^®^, offering a more refined and precise assessment of effusion samples. This methodological distinction is crucial as it allows for a more targeted analysis of cellular components involved in FIP, which is not adequately captured by other routine diagnostic tests, which have been shown to provide only limited diagnostic certainty in FIP. This evaluation is essential because the reagents employed for cell counting in the Sysmex XN-1000V^®^ channels (TNCC-WDF and TNCC-WNR) differed from those used in the analogous channels (DIFF and BASO) of the Sysmex XT-2000iV^®^.

In the present study, the authors presented the first comprehensive analysis of flow cytometry-based ∆TNC using Sysmex XN-1000V^®^ in feline effusions for diagnosing FIP and a novel diagnostic method, the FIP Effusion Index, and its relationship with molecular biology diagnoses.

### 4.5. Recommendations

Future studies should consider incorporating the detection of the FCoV antigen by IHC as part of the diagnostic process. Immunohistochemistry is currently regarded as the gold standard for FIP diagnosis because it directly visualises viral antigens within macrophages in effusion samples and tissue biopsies. While this study focused on comparing ∆TNC analysis and the FIP Effusion Index with molecular biology techniques in effusions, integrating IHC could provide a more comprehensive diagnostic framework. The combination could help to confirm the presence of FCoV in ambiguous cases, thereby reducing false negatives and improving diagnostic accuracy.

Moreover, future research should explore the use of advanced molecular diagnostics, such as next-generation sequencing (NGS), to detect and characterise mutations in the spike protein of the FCoV. These mutations, particularly M1058L and S1060A, are associated with the systemic spread of the virus and are pivotal in the pathogenesis of FIP. By identifying specific mutations that correlate with disease severity and clinical outcomes, NGS could further enhance the diagnostic process and allow for more targeted therapeutic interventions. However, while NGS shows promise in identifying these mutations, they are not exclusive to the disease, which further underscores the relevance of the present study.

Additionally, studies involving a larger and more diverse sample size including cases with septic effusions are recommended. Larger, multi-centre studies involving cats from various geographical locations and different clinical settings would help to validate the robustness and applicability of ∆TNC and the FIP Effusion Index analysis as diagnostic tools. These studies should also aim to standardise the protocols for sample collection, handling, and analysis to minimise variability and improve the reproducibility of results.

Another key area for future investigation is the longitudinal monitoring of ∆TNC and FIP Effusion Index values in monitoring patients with persistent effusions. Tracking these values over time could provide valuable insights into disease progression and response to treatment, potentially identifying new biomarkers that predict clinical outcomes. This approach could also facilitate the development of dynamic diagnostic models that adapt to the changing clinical picture of FIP, enhancing early detection and improving patient management strategies.

In the non-effusive form, given the complexity and diagnostic challenges, there is a pressing need for future research to improve diagnostic accuracy through the integration of advanced methodologies including the combined use of laboratory biomarkers and imaging techniques. Particular emphasis should be placed on developing diagnostic indices that incorporate imaging modalities, such as ultrasonography, alongside biochemical blood markers and/or molecular techniques, such as RT-qPCR on fine-needle aspirates, to establish robust diagnostic correlations.

Furthermore, given the parallels between FIP and coronavirus-induced diseases in humans, such as COVID-19, there is an opportunity to investigate the translational potential of these findings. Studies should consider how the diagnostic tools and methods used for FIP in veterinary medicine could be adapted or inspire novel approaches for diagnosing and managing coronavirus-related diseases in humans. This cross-species research could lead to a better understanding of coronavirus pathogenesis, zoonotic transmission, and potential therapeutic targets.

Finally, it is essential for future research to focus on the development of rapid, point-of-care diagnostic tests for FIP that combine the sensitivity and specificity of laboratory-based assays like ∆TNC and the FIP Effusion Index analysis with the convenience and speed required for routine clinical practice. Such advancements would significantly enhance the ability of veterinarians to diagnose FIP quickly and accurately, facilitating timely treatment and improving the overall prognosis for affected cats.

By addressing these recommendations, future studies can build upon the current findings and contribute to more effective diagnostic strategies for FIP, ultimately improving the health and welfare of cats and potentially offering broader insights into coronavirus infections across species.

## 5. Conclusions

The present study demonstrates that the Sysmex XN-1000V^®^ generated ∆TNC provides exceptionally high diagnostic accuracy for FIP. This accuracy is attributed to the formation of clots in the WNR channel after mixing with the reagent, which trap cells similarly to the mechanism observed in the Rivalta test, a method also noted for its high diagnostic accuracy for FIP. This clotting reaction leads to a low TNCC-WNR, even when the TNCC-WDF counts are high. Therefore, in routine practice, it is not recommended to use the default TNCC counts generated by the WNR channel, but to directly use the TNCC-WDF and especially the ∆TNC, particularly when FIP is suspected. In these cases, a ∆TNC ≥ 2.1 is highly suggestive of FIP, and a ∆TNC ≥ 4.9 may be considered as diagnostic for FIP.

The FIP Effusion Index, which integrates the ALB/GLOB ratio with the ∆TNC, has emerged as an exceptionally robust diagnostic tool. The ALB/GLOB ratio itself provided valuable information about the protein composition of the effusion, with a ratio ≤ 0.56 yielding a sensitivity of 96.3% and a specificity of 95.7% for FIP diagnosis. When combined in the FIP Effusion Index, these parameters enhance diagnostic precision. A FIP Effusion Index value equal to or greater than 5.06 resulted in a sensitivity of 96.3% and specificity of 95.7%, while an optimal cutoff of ≥7.54 achieved perfect specificity (100%) and a sensitivity of 96.3%. This makes the FIP Effusion Index a highly effective and reliable tool for accurately identifying FIP, providing superior diagnostic accuracy compared to the individual use of the ALB/GLOB ratio or ∆TNC. The use of these combined measures should be considered a standard approach in veterinary medicine practice for the diagnosis of FIP, enabling more precise and confident clinical decision-making and ultimately improving patient outcomes.

This research provides valuable insights into the broader context of coronavirus-related diseases, offering potential guidance for diagnostic approaches in other species with effusive forms of coronavirus infections, as suggested by the Stockholm paradigm. This cross-species relevance underscores the importance of this research in enhancing our understanding of coronavirus pathogenesis and improving diagnostic and therapeutic strategies in both veterinary and human medicine.

## Figures and Tables

**Figure 1 vetsci-11-00563-f001:**
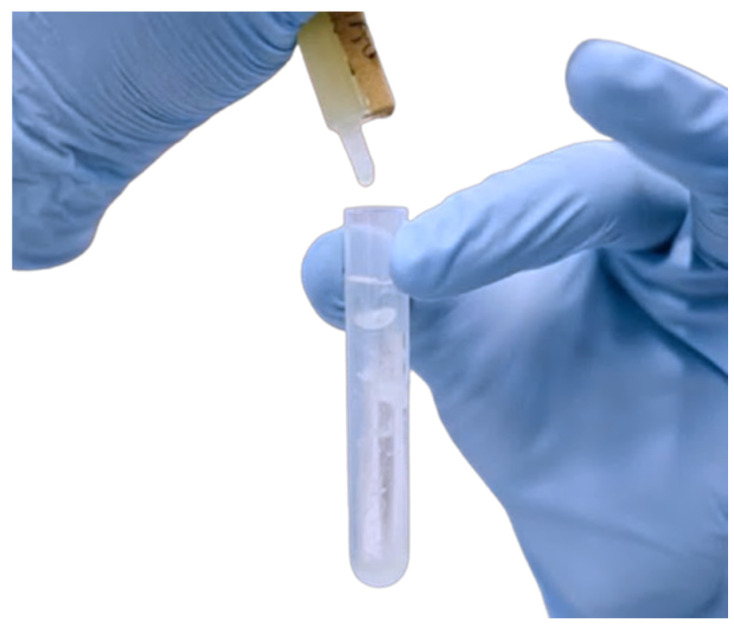
Positive Rivalta test. This is identified by the formation of a precipitate that adheres to the surface, retains its shape, or slowly settles to the bottom of the solution in the test tube. The image shows the precipitate clearly forming within the tube, indicating a positive result for the Rivalta test.

**Figure 2 vetsci-11-00563-f002:**
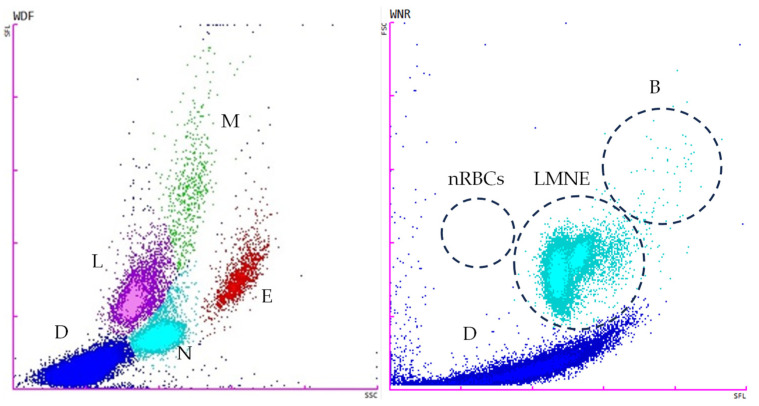
Scattergrams adapted from the Sysmex XN-1000V^®^ in whole blood mode with the EDTA blood feline sample. The figure on the **left** displays the TNCC-WDF (total nucleated cells in the white blood cell differential channel), while the figure on the **right** illustrates the TNCC-WNR (total nucleated cells in the white blood cell nucleated channel). B, basophils; D (blue), debris; E (red), eosinophils; L (purple), lymphocytes; LMNE, all the WBC populations except basophils; M (green), monocytes; N (light blue), neutrophils; nRBCs, nucleated red blood cells.

**Figure 3 vetsci-11-00563-f003:**
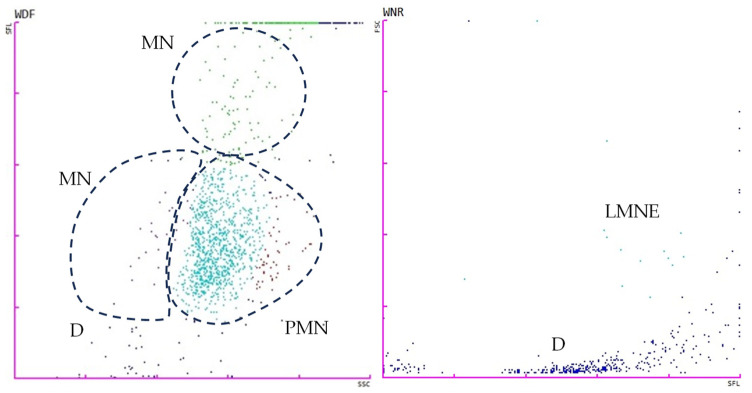
Scattergrams adapted from the Sysmex XN-1000V^®^ in whole blood mode with effusion from a case of feline infectious peritonitis. The figure on the **left** displays the TNCC-WDF (total nucleated cells in the white blood cell differential channel), while the figure on the **right** illustrates the TNCC-WNR (total nucleated cells in the white blood cell nucleated channel). D (blue), debris; LMNE (light blue), all the WBC populations except basophils; MN, mononuclear cells (macrophages, monocytes, lymphocytes); PMN, polymorphonuclear cells (mainly neutrophils). Note that there was a marked reduction in events in the WNR channel.

**Figure 4 vetsci-11-00563-f004:**
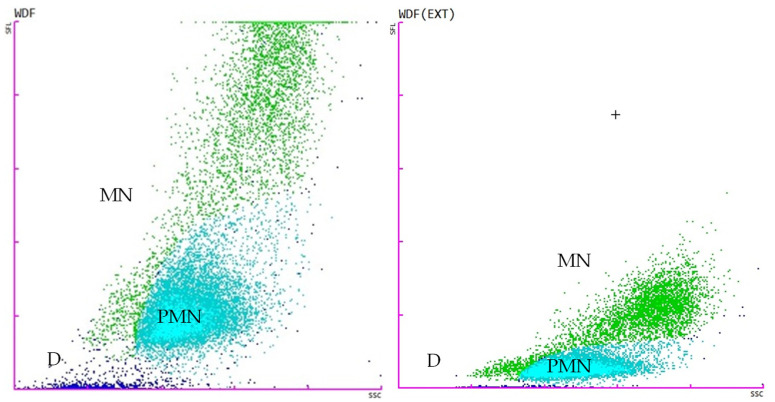
Scattergrams adapted from the Sysmex XN-1000V^®^ in WBC-BF (total white blood cell count in body fluid mode) with effusion from a case of feline infectious peritonitis (FIP). The figure on the **left** displays the WDF-BF, while the figure on the **right** illustrates the WDF(EXT)-BF (extended). D (blue), debris (cell detritus, compound particles, other cells); MN (green), mononuclear cells (macrophages, monocytes, lymphocytes); PMN (light blue), polymorphonuclear cells (mainly neutrophils); +, high fluorescence events (epithelial cells, cell conglomerates).

**Figure 5 vetsci-11-00563-f005:**
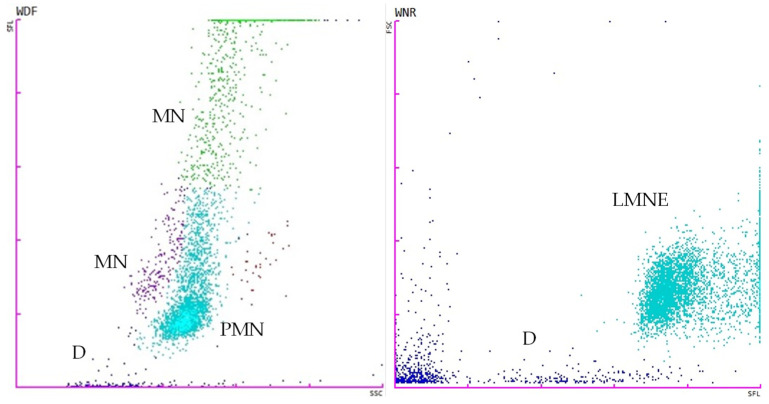
Scattergrams adapted from the Sysmex XN-1000V^®^ in whole blood mode with and effusion from a non-feline infectious peritonitis (FIP) case. The figure on the **left** displays the TNCC-WDF (total nucleated cells in the white blood cell differential channel), while the figure on the **right** illustrates the TNCC-WNR (total nucleated cells in the white blood cell nucleated channel). D (blue), debris; LMNE (light blue), all the WBC populations except basophils; MN (purple and green), mononuclear cells (macrophages, monocytes, lymphocytes); PMN (light blue), polymorphonuclear cells (mainly neutrophils). Note that there was no reduction in events in the WNR channel.

**Figure 6 vetsci-11-00563-f006:**
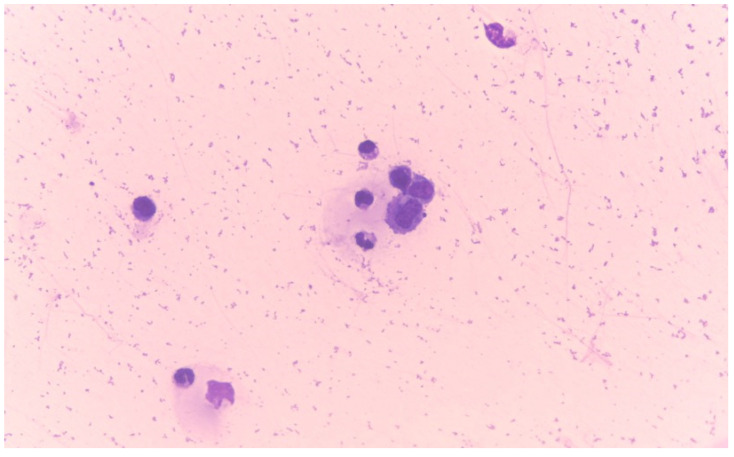
Cytology of a feline infectious peritonitis (FIP) effusion. Mixed cell population (non-degenerate neutrophils and macrophages) dispersed in a granular eosinophilic proteinaceous background commonly observed in FIP effusions. May-Grünwald Giemsa, 400×.

**Figure 7 vetsci-11-00563-f007:**
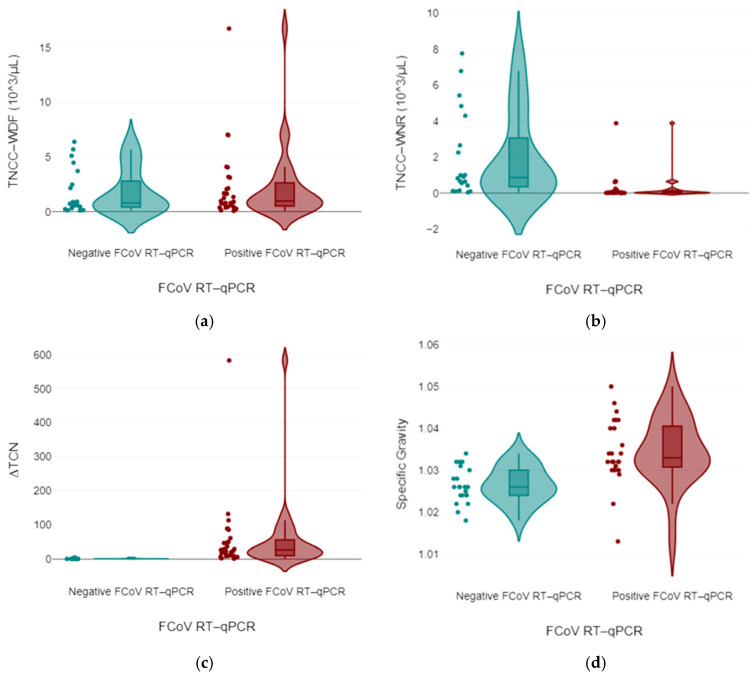
Values of TNCC-WDF (**a**), TNCC-WNR (**b**), ratio between total nucleated cell counts in the WDF and WNR channel of the laser counter Sysmex XN-1000V^®^, reported as ∆WBC by the instrument but termed ∆TNC for this study (**c**), and values of specific gravity (SG) (**d**) recorded in cats with FIP and in cats with diseases other than FIP (non-FIP). The boxes indicate the I–III interquartile range (IQR), and the horizontal line indicates the median. Dots indicate the values recorded in this study. The TNCC-WDF and the TNCC-WNR graphs do not include the results of two non-FIP specimens that had extremely high TNCC-WDF and TNCC-WNR counts (342.5 and 222.6 cells × 10^3^/μL, respectively).

**Figure 8 vetsci-11-00563-f008:**
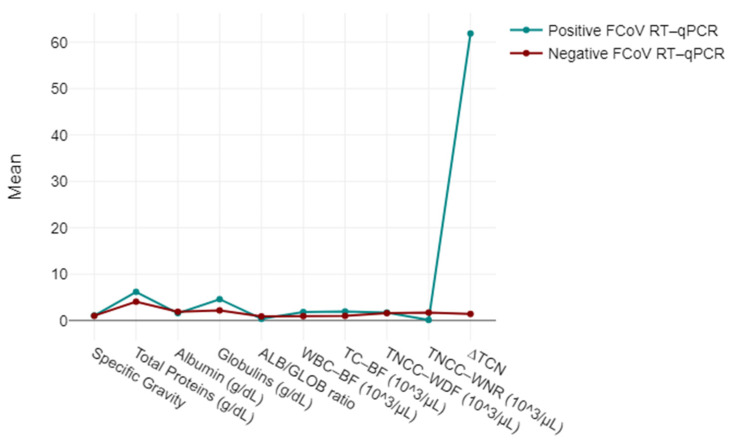
Mixed-model ANOVA findings: biochemical and flow cytometry data, categorising them based on their responses to the Feline Coronavirus RT-qPCR Kit^®^ test as either negative or positive.

**Figure 9 vetsci-11-00563-f009:**
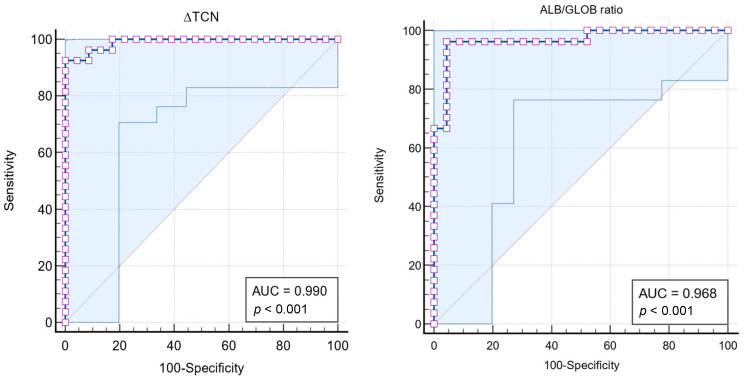
Area under the curve-receiver operating characteristic (AUC-ROC). The figure on the **left** displays the AUC-ROC of ∆TNC in the effusions, total nucleated cell derived from the ratio between TNCC-WDF and TNCC-WNR counts on the Sysmex XN-1000V^®^, showing strong correlation and statistical significance (*p* < 0.001) with the FCoV real-time RT-qPCR test results. The figure on the **right** illustrates the AUC-ROC of the ALB/GLOB ratio in the effusions, showing a strong correlation and statistical significance (*p* < 0.001) with the FCoV real-time RT-qPCR test results.

**Figure 10 vetsci-11-00563-f010:**
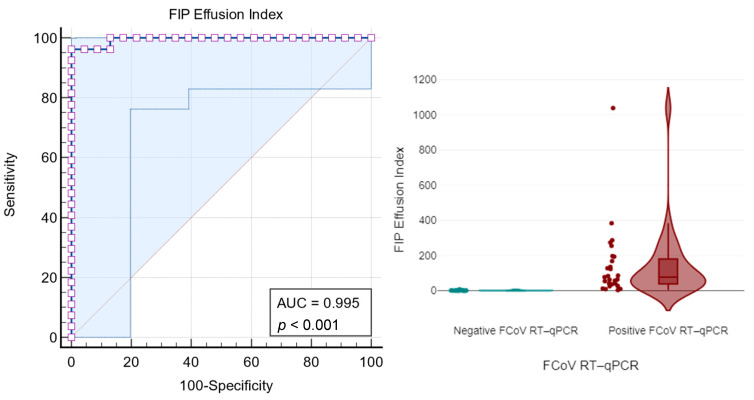
The figure on the **left** displays the area under the curve-receiver operating characteristic (AUC-ROC) of the FIP Effusion Index, showing strong correlation and statistical significance (*p* < 0.001) with the FCoV real-time RT-qPCR test results. The figure on the **right** illustrates the FIP Effusion Index in cats with FIP and cats with diseases other than FIP (non-FIP). The boxes indicate the I–III interquartile range (IQR), and the horizontal line indicates the median. Dots indicate the values recorded in this study.

**Figure 11 vetsci-11-00563-f011:**
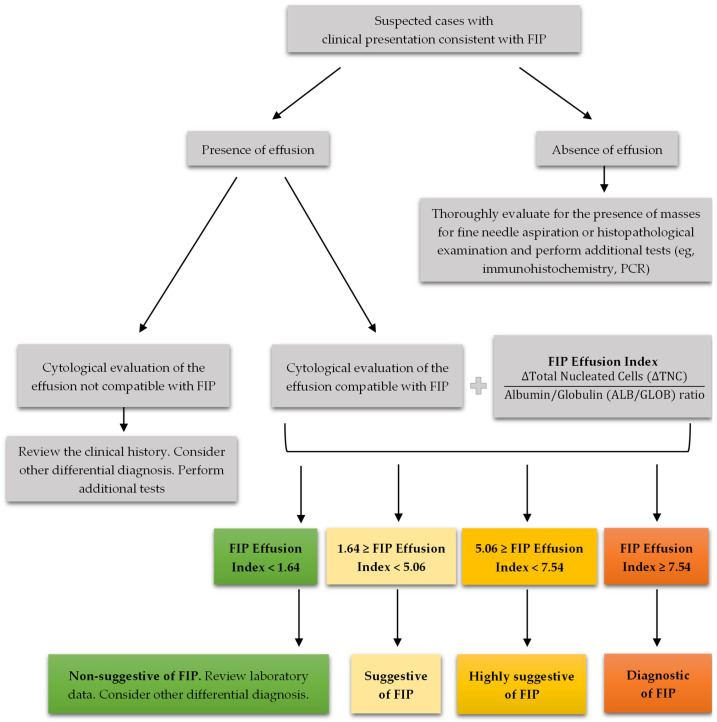
Diagnosis algorithm for evaluating suspected cases of feline infectious peritonitis (FIP) in cats based on their clinical presentation. If effusion is present, the diagnostic process begins with a cytological evaluation of the effusion. If the cytological findings are not compatible with FIP, the clinical history should be reviewed, and other differential diagnoses should be considered. If the effusion is compatible with FIP, the FIP Effusion Index is calculated using the albumin-to-globulin (ALB/GLOB) ratio and the delta total nucleated cell (∆TNC) count. The interpretation of the FIP Effusion Index is as follows: an index value below 1.64 is not suggestive of FIP, and further review of the laboratory data and consideration of other differential diagnoses are recommended. An index value between 1.64 and less than 5.06 is suggestive of FIP; an index value between 5.06 and less than 7.54 is highly suggestive of FIP, whereas an index value equal to or greater than 7.54 is diagnostic of FIP. If there is an absence of effusion, the algorithm advises a thorough evaluation for the presence of masses through fine needle aspiration or histopathological examination, along with additional tests such as immunohistochemistry or PCR. This structured approach helps clinicians accurately differentiate FIP from other conditions, thereby enhancing diagnostic precision and patient management.

**Table 1 vetsci-11-00563-t001:** Traditional Classification Scheme using criteria based on protein levels and cell counts.

Type	Total Protein (g/dL)	Total Cell Count (cells/μL)
Transudate	<2.5	<1500
Modified transudate	2.5–7.5	1000–7000
Exudate	>3.0	>7000

**Table 2 vetsci-11-00563-t002:** Thermal cycling conditions used for the detection of FCoV in the effusion samples.

Cycles	Temperature	Time	Notes
1	50 °C	20 min	Reverse transcription
1	95 °C	2 min	Polymerase activation
40	95 °C	5 s	Denaturation
	60 °C	30 s	Annealing/Extension

**Table 3 vetsci-11-00563-t003:** Negative and positive molecular biology test by sex in the 50 animals included in this study.

	Feline Coronavirus RT-qPCR		
Sex	Negative	Positive	Total
Female	11 (22%)	7 (14%)	18 (36%)
Male	12 (24%)	20 (40%)	32 (64%)
Total	23 (46%)	27 (54%)	50 (100%)

**Table 4 vetsci-11-00563-t004:** Occurrence of coronavirus identification by age in the 50 animals included in this study.

Age Group							
		Kitten (<1 Year Old)	Young (1 to <2 Years Old)	Adult (2 to <6 Years Old)	Senior (6 to <11 Years Old)	Old (≥11 Years Old)	Total
Feline Coronavirus RT-qPCR	Negative	4 (8%)	6 (12%)	5 (10%)	3 (6%)	5 (10%)	23 (46%)
	Positive	9 (18%)	6 (12%)	8 (16%)	4 (8%)	0 (0%)	27 (54%)
	Total	13 (26%)	12 (24%)	13 (26%)	7 (14%)	5 (10%)	50 (100%)

**Table 5 vetsci-11-00563-t005:** Positive and negative Rivalta tests according to the Feline Coronavirus RT-qPCR.

		Rivalta Test				
		Negative		Positive		Total
		*n*	% Within Rivalta Test	*n*	% Within Rivalta Test	*n*
Feline Coronavirus RT-qPCR	Negative	16 (32%)	69.6% (specificity)	7 (14%)	30.4%	23 (46%)
	Positive	1 (2%)	3.7%	26 (52%)	96.3% (sensitivity)	27 (54%)
	Total	17 (34%)		33 (66%)		50 (100%)

**Table 6 vetsci-11-00563-t006:** Effusion type according to Feline Coronavirus RT-qPCR.

		Effusion Type		
		Modified Transudate	Exudate	Total
Feline Coronavirus RT-qPCR	Negative	21 (42%)	2 (4%)	23 (46%)
	Positive	25 (50%)	2 (4%)	27 (54%)
	Total	46 (92%)	4 (8%)	50 (100%)

**Table 7 vetsci-11-00563-t007:** Biochemical and flow cytometry data WBC-BF, TC-BF, TNCC-WDF, TNCC-WNR, and ∆TNC, in effusions, categorised according to the Feline Coronavirus RT-qPCR Kit^®^ test results, showing comparisons between cats with either negative or positive outcomes.

			Feline Coronavirus RT-qPCR in Effusions	
Effusion			Negative	Positive
Parameter (Units)	Minimum	Maximum	Median (IQR)	Median (IQR)
Specific gravity (SG)	1.01	1.05	1.03	1.03
TP (g/dL)	2.80	10.76	4.00	5.64
ALB (g/dL)	1.02	2.47	1.93	1.65
GLOB (g/dL)	1.25	9.11	2.00	4.15
ALB/GLOB ratio	0.18	1.5	0.93	0.35
WBC-BF (10^3^/μL)	0.05	341,612	0.63	1.11
TC-BF (10^3^/μL)	0.05	343,632	0.64	1.12
TNCC-WDF (10^3^/μL)	0.06	335,752	0.88	0.97
TNCC-WNR (10^3^/μL)	0.00	342,445	1.00	0.03
∆TNC	0.30	583	0.93	27.04

TP (g/dL), absolute levels of total proteins; ALB (g/dL), absolute levels of albumin; GLOB (g/dL), absolute levels of globulins; WBC-BF (10^3^/μL), total white blood cell count in body fluid mode; TC-BF (10^3^/μL), total nucleated cells in body fluid mode; TNCC-WDF (10^3^/μL), total nucleated cells in the white blood cell differential channel in whole blood mode; TNCC-WNR (10^3^/μL), total nucleated cells in the white blood cell nucleated channel in whole blood mode; ∆TNC, total nucleated cell derived from the ratio between the TNCC-WDF and TNCC-WNR counts on the Sysmex XN-1000V^®^.

**Table 8 vetsci-11-00563-t008:** Criterion values and coordinates of the ROC curve for ∆TNC.

∆TNC	Sensitivity (%)	95% CI	Specificity (%)	95% CI
≥1.115	96.30	81.0–99.9	86.96	66.4–97.2
≥2.081	96.30	81.0–99.9	91.30	72.0–98.9
≥2.506	92.59	75.7–99.1	91.30	72.0–98.9
≥4.903	92.59	75.7–99.1	100	85.2–100

∆TNC, total nucleated cells derived from the ratio between the TNCC-WDF and TNCC-WNR counts on the Sysmex XN-1000V^®^.

**Table 9 vetsci-11-00563-t009:** Criterion values and coordinates of the ROC curve for the FIP Effusion Index.

FIP Effusion Index	Sensitivity (%)	95% CI	Specificity (%)	95% CI
≥1.637	100	87.2–100	86.96	66.4–97.2
≥4.968	96.30	81.0–99.9	91.30	72.0–98.9
≥5.059	96.30	81.0–99.9	95.65	78.1–99.9
≥7.542	96.30	81.0–99.9	100	85.2–100

## Data Availability

The data presented in this study are available upon request from the corresponding authors.

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
