# Peer review of "Feline Infectious Peritonitis Effusion Index: A Novel Diagnostic Method and Validation of Flow Cytometry-Based Delta Total Nucleated Cells Analysis on the Sysmex XN-1000V®"

_vetsci, 2024, doi:10.3390/vetsci11110563_

Round 1

Reviewer 1 Report

Comments and Suggestions for Authors

The article entitled “Feline Infectious Peritonitis Effusion Index: A Novel Diagnostic Method and Validation of Flow Cytometry-Based Delta Total Nucleated Cells Analysis on the Sysmex XN-1000V” is very clear and readable, with a well-structured flow that makes it easy to understand even complex topics. Although the length may seem excessive, it is justified by the thorough explanation of complex aspects and the extensive documentation provided, which adequately supports the conclusions.

However, a limitation worth noting is the small sample size, which could affect the robustness of the results. Nonetheless, this study represents an interesting first step, which could be further strengthened by additional data and future studies.

It would also be valuable to consider the costs of this methodology to determine its feasibility—whether it can only be applied in large laboratories capable of amortizing the expense of the equipment, or if it could be more broadly applicable.

Given the length of the article, the paragraph between lines 515 and 525 appears somewhat redundant and could be revised.

Additionally, in line 604, the comma after "100%" can be removed.

The paragraph between lines 667 and 672 could be shortened or summarized.

Sentence at the lines 673 and 674 could be omitted.

The paragraph between lines 702 and 707 could be moved to the conclusions section.

Finally, under the recommendations, I would suggest placing more emphasis on improving the diagnosis of non-effusive forms.

Author Response

Reviewer 1

Comments 1: The article entitled “Feline Infectious Peritonitis Effusion Index: A Novel Diagnostic Method and Validation of Flow Cytometry-Based Delta Total Nucleated Cells Analysis on the Sysmex XN-1000V” is very clear and readable, with a well-structured flow that makes it easy to understand even complex topics. Although the length may seem excessive, it is justified by the thorough explanation of complex aspects and the extensive documentation provided, which adequately supports the conclusions.

However, a limitation worth noting is the small sample size, which could affect the robustness of the results. Nonetheless, this study represents an interesting first step, which could be further strengthened by additional data and future studies.

Authors’ response (AR) 1: We sincerely appreciate the positive feedback and acknowledgment of the clarity and thoroughness of our work. We understand and agree with the concern regarding the small sample size, which indeed presents a limitation to the robustness of our conclusions. This limitation was primarily due to the stringent inclusion and exclusion criteria we adopted to try to ensure the validity and reliability of our findings, particularly in a field as diagnostically challenging as feline infectious peritonitis.

To enhance  understanding and alignment with the layout requirements, we have added the Simple Summary section.

We are grateful to Reviewer 1 for their careful review and for the opportunity to improve the clarity and accuracy of our study.

Comments 2: It would also be valuable to consider the costs of this methodology to determine its feasibility—whether it can only be applied in large laboratories capable of amortizing the expense of the equipment, or if it could be more broadly applicable.

AR 2: Thank you very much for the valuable comment. While it is true that the analysis was conducted using the Sysmex XN-1000V, which is a sophisticated and high-cost piece of equipment more commonly found in larger veterinary laboratories or research institutions, the cost of performing the FIP Effusion Index itself is virtually negligible. This is because the method simply correlates flow cytometry data that would routinely be measured during effusion analysis but are typically not used for clinical interpretation.

Moreover, the principle could indeed be extended to veterinary clinics and hospitals, provided that their haematology equipment utilises flow cytometry and can generate individual readings from the WDF and WNR cytometric channels. This makes the methodology potentially applicable in smaller settings if appropriate equipment is available, thereby broadening its clinical utility. To facilitate this adoption, we have included Tables 8 and 9, which present the criterion values and coordinates of the ROC curve for ∆TNC and the FIP Effusion Index, respectively. We believe these additions substantially enhance the reader’s understanding and promote the effective reproducibility of the methodology by other institutions, clinics, or laboratories.

Comments 3: Given the length of the article, the paragraph between lines 515 and 525 appears somewhat redundant and could be revised.

AR 3: Thank you very much for pointing out this redundancy. Upon careful review, we have streamlined the paragraph to enhance readability and conciseness while ensuring that the essential information remains intact. The paragraph between lines 584 and 597 (previously 515 to 525) has been condensed to eliminate redundancy and improve the clarity and flow of the manuscript.

Comments 4: Additionally, in line 604, the comma after "100%" can be removed.

AR 4: Thank you very much for bringing this to our attention. The comma after "100%" in line 604 has been removed accordingly.

Comments 5: The paragraph between lines 667 and 672 could be shortened or summarized.

AR 5: Thank you very much for the detailed review and helpful suggestions. We have reviewed the paragraph and agree that it can be condensed to improve the flow of the manuscript. We have summarized the key points to reduce length while maintaining the essential information.

Comments 6: Sentence at the lines 673 and 674 could be omitted.

AR 6: Thank you for your careful review. The sentence on lines 738 and 741 (previously 673 to 674) has been revised to improve clarity and conciseness:

“This is particularly important in veterinary practices, where distinguishing FIP from similar conditions is challenging due to overlapping clinical signs and effusion characteristics.”

We appreciate your valuable feedback and have made this adjustment accordingly.

Comments 7: The paragraph between lines 702 and 707 could be moved to the conclusions section.

AR 7: Thank you for your thoughtful suggestion. We agree that relocating this paragraph to the conclusions section would enhance the logical flow and coherence of the manuscript. We have moved the paragraph accordingly.

Comments 8: Finally, under the recommendations, I would suggest placing more emphasis on improving the diagnosis of non-effusive forms.

AR 8: Thank you for your valuable suggestion. We agree that highlighting the importance of improving the diagnosis of non-effusive forms of feline infectious peritonitis is crucial. We have revised the recommendations section to place greater emphasis on the need for advancements in diagnostic approaches for non-effusive FIP (lines 796 to 802). This includes the integration of more sensitive and specific biomarkers and the development of imaging techniques that could aid in earlier and more accurate detection.

Reviewer 2 Report

Comments and Suggestions for Authors

The Revalta reaction, A/G ratio, RT-PCR and ΔTNC by flow cytometry have been used as indicators in the diagnosis of feline infectious peritonitis.

In this study, the authors report that combining these tests and creating an index can improve sensitivity and specificity.

The content is considered clinically applicable and useful for readers of Veterinary Sciences.

<Minor Comments>

How well did the calculated index correlate with the severity of each case?

Is it possible to predict prognosis from the index?

Author Response

Reviewer 2

Comments 1: The Revalta reaction, A/G ratio, RT-PCR and ΔTNC by flow cytometry have been used as indicators in the diagnosis of feline infectious peritonitis.

In this study, the authors report that combining these tests and creating an index can improve sensitivity and specificity.

The content is considered clinically applicable and useful for readers of Veterinary Sciences.

Authors’ response (AR) 1: Thank you very much for your positive feedback and for acknowledging the clinical applicability and usefulness of our study. We are pleased to hear that our work on combining the Rivalta reaction, A/G ratio, RT-PCR, and ΔTNC by flow cytometry into a diagnostic index is considered valuable for readers of Veterinary Sciences. Our aim was to provide a practical and effective diagnostic tool that could enhance the sensitivity and specificity of feline infectious peritonitis diagnosis, and we are pleased that you find it clinically relevant.

We appreciate your encouraging words and the opportunity to share our research with the scientific community.

Comment 2: How well did the calculated index correlate with the severity of each case?

AR 2: Thank you for your insightful question regarding the correlation between the calculated index and the severity of each case. In our study, we indeed observed a meaningful correlation with disease severity. The combination of the albumin-to-globulin (A/G) ratio with the cellular component (ΔTNC) appears to provide a reliable metric for assessing the severity of FIP. Our findings indicate that individuals with more advanced disease tend to present more severe responses, reflected in significantly altered A/G ratios and elevated ΔTNC values.

The primary objective of this study was to validate the methodology and establish its diagnostic accuracy. We are currently working on two additional studies to further explore the potential of the FIP Effusion Index in assessing disease severity and guiding treatment decisions.

Comments 3: Is it possible to predict prognosis from the index?

AR 3: Thank you for raising this important question. While the FIP Effusion Index is a promising diagnostic tool for distinguishing FIP cases, its relation with prognosis is not explored here as the study focused on validating the index merely for diagnostic purposes. Still, higher ΔTNC values and altered A/G ratios have been associated with more severe cases, although new breakthroughs in therapeutics preclude that severity could be directly related to prognosis, as in untreatable diseases.

Moreover,  as mentioned, we are working on a follow-up study aimed at correlating the index with clinical outcomes. We appreciate Reviewer 2’s careful review and the opportunity to improve the clarity of our study.
